# Evaluation of an Integrated Ultrafiltration/Solid Phase Extraction Process for Purification of Oligomeric Grape Seed Procyanidins

**DOI:** 10.3390/membranes10070147

**Published:** 2020-07-09

**Authors:** Alba Gutierrez-Docio, Paula Almodóvar, Silvia Moreno-Fernandez, Jose Manuel Silvan, Adolfo J. Martinez-Rodriguez, Gonzalo Luis Alonso, Marin Prodanov

**Affiliations:** 1Functional Food Ingredients Group, Department of Production and Characterization of Novel Foods, Instituto de Investigación en Ciencias de la Alimentación (CIAL), (CEI, CSIC-UAM), C/Nicolás Cabrera, 9. Campus Cantoblanco, Universidad Autónoma de Madrid, 28049 Madrid, Spain; alba.gutierrez@uam.es (A.G.-D.); palmodovar@pharmactive.eu (P.A.); silvia.moreno@uam.es (S.M.-F.); 2Pharmactive Biotech Products SL, Parque Científico de Madrid, 28049 Madrid, Spain; 3Microbiology and Food Biocatalysis Group, Department of Biotechnology and Food Microbiology, Instituto de Investigación en Ciencias de la Alimentación (CIAL) (CEI, CSIC-UAM), C/ Nicolás Cabrera, 9. Campus Cantoblanco, Universidad Autónoma de Madrid, 28049 Madrid, Spain; jm.silvan@csic.es (J.M.S.); adolfo.martinez@csic.es (A.J.M.-R.); 4Cátedra de Química Agrícola, E.T.S.I. Agrónomos y Montes, Universidad de Castilla-La Mancha, Avda. de España s/n, 02071 Albacete, Spain; Gonzalo.Alonso@uclm.es

**Keywords:** grape seed extract, purification, tangential-flow pressure-driven membrane ultrafiltration, solid-phase extraction, oligomeric procyanidins

## Abstract

The effectiveness of a preparative integrated ultrafiltration/solid-phase extraction (UF/SPE) process for purification of oligomeric procyanidins (OPCs) from a crude grape seed extract (GSE) was studied for the first time. The separation of OPCs from polymeric procyanidins (PPCs) by UF was very efficient. The membrane showed an acceptable filtration flux of 6 to 3.5 L/h·m^2^ at 0.5 bar of transmembrane pressure and 95% recovery of its water flux after chemical cleaning. The process was scalable to a pilot scale. The separation of very polar and ionic species from OPCs by SPE (XAD7HP and XAD16 resins) was also very good, but both adsorbents lost their retention capacities quickly, due probably to irreversible retention of OPCs/PPCs. Even though the global purification of OPCs by the integrated UF/SPE process allowed the recovery of 24.2 g of highly purified OPCs (83% purity) from 14.4 L of crude grape seed extract, the use of these adsorbents for further purification of the OPCs was very limited.

## 1. Introduction

Flavanols (catechins and procyanidins) are the most abundant secondary metabolites found in seeds of *Vitis vinifera* L. species with amounts of up to 6.4 g/100 g [1]. From a chemical point of view, they are an excellent example of how plants can synthesise an unlimited diversity of structures based on the condensation of only three elemental units: (+)-catechin, (−)-epicatechin and (−)-epigallocatechin. This diversity is due to some of their particular features: the stereochemistry of the asymmetric carbon atoms C2 and C3 of the flavan skeleton, type of interflavan bond (C4–C8 and C4–C6 for B-type procyanidins), the length of the polymer chain (degree of polymerisation), the degree of galloylation and the position of the gallic acid ester [2,3,4] and its ability to form complex structures with other biopolymers, such as polysaccharides and proteins [5]. 

From a technological point of view, procyanidins grant bitter taste and astringency to red wines [6] and contribute to the stabilisation of their colour and equilibration of their redox state [7]. From a nutritional point of view, they have been considered as antinutrients for a long time because of their capacity to inhibit digestive enzymes [8]. Nowadays, a growing number of studies show that they are rather beneficial for animal [9,10,11] and human health, covering a large number of activities such as cardioprotective [12], anticarcinogenic [13], or antiaging [14], among others. Most of these activities have been attributed to the high reducing (antioxidant) activity of the procyanidins [15], which is, respectively, 20- and 50-times higher than those of the most studied natural antioxidants, vitamins C and E [16]. Besides this, grape seed procyanidins have demonstrated huge antimicrobial [17,18], antiulcer [19], anti-inflamatory [15], or antilithiasis [20] activities, among others. 

Nevertheless, it should be pointed out that only monomeric to trimeric flavan-3-ols have been found to cross animal and human intestinal walls at trace concentrations [21,22,23,24], while procyanidins, with a higher degree of polymerisation, are not absorbed or, rather, are partially hydrolysed to monomers and dimers in the stomach [25] or transformed to more simple metabolites by gut microbiota [24,26,27]. Some authors have also claimed that the antioxidant activity of procyanidins rises along with the increase in their degree of polymerisation, up to trimers, and afterwards, decreases [28]. All these findings suggest that from a health and nutritional point of view, low molecular mass oligomeric procyanidins (OPCs) are more interesting than those of a higher degree of polymerisation. 

The vast structural diversity and content variability of grape seed procyanidins make their separation and determination extremely difficult. In fact, there is not a singular analytical method that can resolve this problem completely. Due to their high to moderate molecular polarities, procyanidins are individually assessed using reverse-phase liquid chromatography. Nevertheless, in spite of the important technical advances achieved in chromatographic separations, such as modern high-performance liquid chromatographs coupled to photodiode arrays and high-resolution mass spectrometry detectors (HPLC–PAD–MS), ultra HPLC–PAD–MS (UPLC–MS), or two-dimensional HPLC–MS, only some major procyanidins of up to heptamers have been separated and, up to tetramers, have been quantified [24,29,30,31]. 

For global characterisation of monomer and polymer flavan-3-ols, colourimetric methods are usually used based on their acid depolymerisation in the presence of butanol, condensation with aldehydes, or reaction with specific reagents (Folin–Ciocalteu method), among others [29]. The main problem with these methods is that they have relatively low sensitivity and reproducibility that lead to considerable variations between methods and laboratories. To improve the limitations of these methods, other strategies could be of choice. In this sense, normal-phase (NP) HPLC seems more suitable. This mode of separation is usually used for the analysis of molecules with low polarities but was adapted for the analysis of procyanidins by Rigaud et al. [32]. In such a way, procyanidins elute in order of growing molecular masses and, in the case of cocoa procyanidins, could be quite easy to resolve and quantify, mainly due to the simplicity of their structures (which are based only on the elemental unit (-)-epicatechin). It was later understood that, at this mode of separation, procyanidins precipitate on the head of the column, as they are introduced when the less polar component of the mobile phase is pumped through the column (in which they are not or are slightly soluble), and afterwards, they redissolve gradually with the increase of the more polar component [33], where first, the molecules with lower masses pass in solution and then the bigger ones. Nevertheless, applying NP–HPLC to the analysis of grape seed procyanidins can resolve only procyanidins up to trimers. Separation of higher oligomers is much more difficult because of the widespread overlapping of the galloylated forms of procyanidins with a lower degree of polymerisation on the nongalloylated procyanidins with a higher degree of polymerisation. Therefore, this mode of analysis does not allow a fine separation of grape seed OPCs but does allow the depiction of their profile with an increasing order of molecular masses, as well as the separation of higher molecular mass polymers in a relatively singular peak at the end of the chromatogram, making an objective evaluation of the proportion between monomers, oligomers and polymers [34]. 

Preparative separation of grape seed procyanidins is still more difficult and includes several purification stages in which the number sequence is conditioned by the proposed purity requirements. For initial purification, usually, more unspecific techniques are adopted, such as SEC on relatively inert materials (Sephadex or Toyopearl) [35,36,37], solvent/nonsolvent precipitation (SNSP) of highly polymerised procyanidins [38] or SPE, preferably performed with synthetic macroporous adsorbents [39]. An alternative to SEC and SNSP is pressure-driven tangential-flow ultrafiltration (UF) that allows relatively good and fast separation of macromolecules, such as procyanidins with a high degree of polymerisation from smaller molecules, including OPCs. Nevertheless, there is only one study in the literature regarding the separation of grape seed procyanidins by UF, with very limited use because it was carried out at an analytical scale [40]. For finer separation, chromatographic techniques with greater resolution capacities are more appropriate. The most important of them are counter-current chromatography [38,41,42] and low/middle-pressure liquid chromatography [35,43]. However, it should be pointed out that these operations are very expensive, and they are used, rather, for obtaining relatively small amounts of procyanidins for testing their biological properties or for analytical reference substances. 

Thus, the present study aims to explore the effectiveness of a preparative integrated UF/SPE separation process for obtaining an OPC-rich fraction in the order of grams and possible scaling up to a pilot scale.

## 2. Materials and Methods 

### 2.1. Materials and Reagents

The starting material for this work was a concentrated, industrially produced crude grape seed extract (from Output Trade S.L., Villafranca del Penedés, Spain) with 288 NTU of turbidity and 26% of total soluble substances (TSSs). 

For procyanidin purification by UF and SPE by diafiltration, demineralised particle-free water with electrical conductivity of 7–8 µS/cm was obtained inhouse by a Genius 300 reverse osmosis unit (Filtec Depuradoras, Girona, Spain). For HPLC analysis, Milli-Q grade water was produced inhouse by a Milli-Q^®^ Integral 3 purification system (Merck Millipore, Burlington, MA, USA). HPLC grade methanol and acetonitrile were purchased from Scharlab (Barcelona, Spain). Glacial acetic acid (p.a.) was obtained from Sigma-Aldrich (Madrid, Spain) and technical grade acetone and ethanol (96%) from PanReac AppliChem ITW Reagents (Barcelona, Spain). 

### 2.2. Equipment

The crude grape seed extract was clarified by a multifuge 3SR+ centrifuge (Heraeus, SA, Madrid). 

The separation of monomeric and oligomeric (OPC) flavan-3-ols from polymeric procyanidins (PPCs) was carried out on a spiral-wound UF membrane cartridge (model Prep/scale-TFF-6, Merck Millipore) with the following technical characteristics: regenerated cellulose, 5.8 cm in diameter, 39.9 cm in length, 0.53 mm of feed spacer, 0.54 m^2^ of filtration surface and a 10-kDa molecular mass cut-off (MMCO). The membrane cartridge was connected to a pressure-driven tangential-flow UF unit, as described in Muñoz-Labrador et al. [34]. The UF system was operated without external backpressure at a feed flow of 300 L/h, which gave a tangential flow velocity of 3.1 m/s and transmembrane pressure (P_TM_) of 0.5 bar.

The separation of monomeric and OPC flavan-3-ols from sugars, sugar alcohols, di- and tricarboxylic acids, minerals and other non-adsorptive species (hereinafter referred to as sugars) was carried out on a preparative solid-phase extraction (SPE) unit consisting of a 2-L cylindrical glass tube, provided with a glass frit at the bottom for retaining the stationary phase and a Teflon valve for controlling the mobile phase flow speed. An amount of 1 L of XAD7HP (macroreticular aliphatic acrylic cross-linked polymer) resin from Rohm & Haas (Philadelphia, PA, USA) was suspended in 1 L of demineralised particle-free water and packed in the column under a constant flow of 150 mL/min. The resin bed was fixed on the upper side by a 5-mm thick nylon porous tissue (sponge). 

### 2.3. Preparative Separation Methodologies

Clarification of the crude grape seed extract was carried out by centrifugation (Figure 1) by lots of 1.5 L at 4000 rpm (8570× *g*) for 20 min. Turbidity of 0.8 NTU was achieved. The obtained pellet was discarded. The clarified extract was next diluted with demineralised water to 8 g/100 mL of total soluble substances (TSSs; 15 L) for the following UF treatment.

An aliquot of 7.2 L of clarified diluted grape seed extract (CGSE) was ultrafiltered in a continuous concentration mode, at a constant P_TM_ of 0.5 bar, without the termostatisation of the system, as described by Silvan et al. [44]. Initially, the UF system was run with 4 L of CGSE. When 2 L of filtrate was recovered, 2 L of new CGSE was added by the use of a peristaltic pump. At the third load, 1.2 L of new CGSE was added. Amounts of 5.2 and 2 L of permeate and concentrate were obtained, respectively. Monomeric flavan-3-ols and OPCs were recovered with the filtrate. After concentration, purification of the PPCs from OPCs and other small molecules was carried out by diafiltration of the 2 L of concentrate with the addition of 2 L of demineralised water in 3 cycles of dilution/concentration (a total of 6 L of water). The purified PPC fraction (Figure 1) was freeze-dried and stored in the dark at 4 °C. After chemical cleaning of the membrane, a replicate UF with another 7.2 L of GSE was carried out in the same way. 

After filtration, the membrane was cleaned by several washes with demineralised particle-free water, followed by recirculation of 0.1 M NaOH at 45 °C for 60 min and the addition of 200 ppm of sodium hypochlorite (NaClO) for an additional 15 min. 

Membrane hydraulic permeability (L_p_) was determined before (L^0^_p_) and after chemical cleaning (L^C^_p_) by plotting the corresponding water flux values, measured at the feed flow-rate of 660 L/h and 20 °C versus the applied P_TM_, as described previously [45,46].

Monomeric flavan-3-ols and OPCs were purified from sugar alcohols, di- and tricarboxylic acids, minerals, and other very polar and ionic species by preparative solid-phase extraction (SPE) on 1 L column of XAD7HP adsorption resin (Rohm & Haas, Philadelphia, USA). The resin was first conditioned by washing with 2 bed volumes (2 L) of acetone and 5 bed volumes (5 L) of demineralised water. An aliquot of 2.5 L of ultrafiltered grape seed extract (4.2 g/100 mL of TSS) was loaded to the resin at a flow rate of 60 mL/min. Permeate was recovered as a fraction named “sugars”. The retentate was diafiltered with demineralised water until 0 g/100 mL TSS was measured (by refractometry) at the exit of the column. Phenolic compounds were desorbed with 96% ethanol (C+OPC). The solvent was recovered by distillation at reduced pressure, and the dry extract was redissolved with 50–100 mL of water, freeze-dried and stored in the dark at 4 °C.

After each separation set, the resin was regenerated with 2 bed volumes (2 L) of acetone and 5 bed volumes (5 L) of demineralised water. After cleaning, another aliquot of 2.5 L of PPC-free GSE was treated in the same way. The procedure was repeated 4 times until all the ultrafiltrate volume was treated.

### 2.4. Analytical Methodologies

Total soluble substances (TSSs) in water (°Brix) were measured by the Atago handheld refractometer, calibrated to the interval of 0 to 32 g/100 mL of TSSs.

PPCs were determined semiquantitatively by NP–HPLC [33,34]. For this, relative proportions of OPCs and PPCs were calculated by subtracting the joint area of all oligomeric peaks from those of the whole chromatogram (from the peak corresponding to PC_2_G to the end), assuming that PPCs are the major component of the polymeric fraction that absorb at 280 nm. Purified OPC-rich extract from cocoa was used as a complex reference sample for the identification of nongalloylated OPCs.

Mass transference (M_trans_) through the studied membrane was calculated for each measured parameter according to M_trans_ = C_p_/C_f_.100 (%), where C_p_ and C_f_ are the concentration of each parameter in the permeate and feed streams, respectively.

## 3. Results and Discussion

### 3.1. Semiquantitative Determination of OPCs and PPCs by NP–HPLC–PAD

The clarified grape seed extract was analysed by NP–HPLC–PAD. As mentioned above, NP–HPLC does not offer a fine separation of OPCs but does allow us to depict their profile of increasing order of molecular masses, as well as to separate higher polymers in the relatively singular peak at the end of the chromatogram (Figure 4). “Relatively singular peak” comes from the difficult-to-define beginning of the PPC peak because of the existence of an intermedium zone of coelution of oligomers and higher polymers. The use of a cocoa OPC reference material enabled the localization of up to hexamer grape seed nongalloylated OPC peaks. Only those corresponding to nongalloylated and monogalloylated dimers and mono- and digalloylated trimers showed relatively singular resolution (data not shown). Nongalloylated trimers overlapped, most probably with a digalloylated dimer, whereas nongalloylated tetramers overlapped with digalloylated trimers. The rest of the peaks were mixtures of higher nongalloylated and galloylated OPCs. Nevertheless, and in spite of all these limitations, a NP–HPLC chromatogram allows the establishment of a first order of approximation of the relative proportions of OPCs and PPCs.

### 3.2. Separation of Low Molecular Mass (OPCs) from Highly Polymerised Procyanidins (PPCs) by Tangential-Flow Pressure-Driven Ultrafiltration (UF)

The separation of OPCs from PPCs was carried out by tangential-flow UF. A 10-kDa MMCO membrane from regenerated cellulose was selected according to our previous experience (not published data). Before starting UF, a test of the extract permeability throughout the membrane was carried at P_TM_ of 0.25 to 1.0 bar. The response of the permeate flux was linear throughout the whole studied interval. The transference of TSSs throughout the membrane was relatively high, showing a small decrease from 52% to 47% with the increase in pressure. Thus, a moderate P_TM_ of 0.5 bar was chosen, as it corresponded to an acceptable, expected filtration flux of 6 L/hm^2^, a reasonable operation time of 6 to 7 h and a minor risk for membrane fouling.

#### 3.2.1. Effect of the UF on the Main Process Parameters

Two volumes of 7.2 L of GSE were concentrated to 2 L (a concentration factor of 3.6) and 2 lots of 5.2 L of ultrafiltrate were obtained in 178 and 168 min of operation, respectively (Figure 2). The filtration flux decreased from an average value of 6 to 3.5 L/h.m^2^, which corresponds to a 42% loss compared to the initial flux. The slight periodical rises of the flux were due to the slight dilution of the concentrate with the addition of fresh extract (1–2 L).

Both concentrates (2 L of each one) were diluted, respectively, with 2 L of demineralised water and concentrated again to 2 L. This dilution/concentration procedure (diafiltration) was repeated 2 times more to separate the main part of the OPCs and the other molecules with low masses from the retained macromolecular fraction (PPCs) in an additional 170 min. Only 3 cycles of dilution/concentration were carried out in order to minimise oxidation and loss of polymers. The periodical rises of the filtration flux during diafiltration (Figure 2) were due to the dilution of the concentrate with the addition of new volumes of demineralised water (2 L).

Test of the membrane’s hydraulic permeability after standard and forced chemical cleaning showed up to 5% irreversible loss of permeability.

#### 3.2.2. Mass Transfer through the Membrane

Global mass transfer through the 10-kDa MMCO membrane was assessed by handheld refractometer, which allows an easy and fast measurement of the distribution of TSSs into both the permeate and the retentate streams during the whole treatment. The results of this assessment are shown in Figure 3.

At the beginning of UF, 4.2 of the 8 g/100 mL of the TSS content of the initial GSE (47.5%) passed into the permeate flow (Figure 3). The remaining high molecular mass TSSs were concentrated in the retentate stream, reaching an average value of 16.3 g/100 mL at the end of the concentration phase, which is an increase of more than 100%. The small periodical decreases of TSS content in the concentrate flow were due to the dilutions produced by the addition of fresh extract. Contrary to this, the TSS content in the permeate flow showed a very low increase from 4 to 4.6 g/100 mL, which is a direct result of their continuous increase in the retentate stream. These results also indicate a global decrease in mass transfer (TSStrans) during the concentration phase, from 50% to 28%. This decrease could be attributed mainly to the increase of the TSS content in the concentrate; however, the increase in temperature from 15 to 26 °C should also be considered.

The diafiltration process produced an overall decrease of the TSS content in both permeate and concentrate flows due to the continuous removal of low molecular mass species with the permeate. The TSS content of the concentrate decreased from 16.3 g/100 mL at the end of the concentration phase to 9.6 g/100 mL at the end of the diafiltration, while those of the permeate decreased from 4.6 to 1.2 g/100 mL. The periodical decreases of the TSS content of the concentrate were due to the dilution of the concentrate by the addition of water. The mass transfer showed still more marked decreases, reaching values of 6–13% at the end of the diafiltration of both replicates (Figure 3). In this case, this effect should be related mainly to the overall decrease of TSS concentration caused by the dilution with water, which also diminishes the effect of the concentration after water loadings and the temperature increase.

Quantitative mass balance of the UF showed that 14.4 L of clarified grape seed extract with 8 g/100 mL of TSSs was transformed into 10 L of ultrafiltrate with 4.2 g/100 mL of TSSs and 4 L (2 × 2 L) of purified macromolecules (diafiltered concentrate; 0.4 L were stored for analytical needs). This means that, theoretically, 47.5% of the whole dry matter of the extract corresponds to macromolecules. Nevertheless, freeze-drying of the diafiltered concentrate gave 106.1 g of dry powder, which means a recovery of just 20%. This result suggests that diafiltration with only 3 cycles of dilution/concentration with water produced an 80% loss of macromolecules and that this loss will be higher if higher purity is required.

Afterwards, the mass transfer of catechins and procyanidins through the membrane was assessed by NP–HPLC–PAD. Figure 4 shows an overlay of the chromatograms of CGSE, the final permeate and the final concentrate after GSE UF.

As it can be seen, at the end of the concentration phase of the UF (the final concentrate of Figure 4, corresponding to min 166 of Figure 2 and Figure 3), 29%, 38%, 52%, 71% and 88% of catechins, procyanidin dimers, trimers + a digaloylated dimer, tetramers + a trigaloylated trimer and polymers, respectively, were retained in the concentrate stream. This means that the complete separation of oligomeric procyanidins from polymers with this membrane is not possible.

#### 3.2.3. Quality of Separation between Procyanidin Polymers (PPCs) and Oligomers (OPCs)

For quality evaluation of the separation of OPCs from PPCs, freeze-dried fractions of the initial clarified GSE, the whole permeate, the concentrate and the diafiltered concentrate were prepared in a concentration of 20 mg/mL and analyzed by NP–HPLC (Figure 5).

Figure 5 shows that the 10-kDa membrane produced an ultrafiltrate enriched in gallic acid, catechins and procyanidin oligomers of up to tetramers and impoverished in polymers (down to 17 g/100 mL of total PPCs). On the other hand, the diafiltration produced an enrichment of polymers in the retentate (PPCs) of almost 300% but also left other low molecular mass compounds (gallic acid, monomeric (catechins) and oligomeric flavan-3-ols, among others) of an average of 15% in this fraction as impurities [34]. The application of a longer diafiltration process should decrease the amount of these, but the treatment was stopped at this point in order to minimise further losses of PPCs with the diafiltration water. These results suggest that the 10-kDa UF membrane could be used in the purification of catechins and low molecular mass procyanidins (up to tetramers) from higher polymer procyanidins, in addition to the purification of polymer procyanidins from low molecular mass catechins and procyanidin oligomers, assuming the presence of some impurities of polymeric procyanidins in the OPC-rich fraction, as well as some impurities of catechins and oligomeric procyanidins in the PPC-rich fraction.

### 3.3. Separation of OPCs from Sugars by Solid-Phase Extraction (SPE)

The ultrafiltered GSE, containing 4.2 g/100 mL of TSSs, was further submitted to SPE with 1 L of resin placed in a column. The treatment was carried out in 4 batches of 2.5 L/batch. Each permeate was concentrated at reduced pressure to approximately 200 mL and then freeze-dried. Amounts of 58.9 to 60.3 g of dry matter were obtained (Table 1), which correspond to a 56.1–57.4% recovery of molecules free-passing through the adsorbent, mainly sugars and minerals. The OPC-rich fractions obtained after desorption with ethanol were also submitted to distillation for ethanol recovery. The dry residues were dissolved in water and freeze-dried to obtain fluid powders. Amounts of 12.7, 11.5, 10.9 and 10.2 g of dry matter were obtained, respectively, for each batch. Recovery of these fractions was calculated by subtracting the amounts of dry matter entering the resin (per batch) and the amounts recovered in the permeates. Thus, these data were much lower than those obtained for the permeate constituents, ranging in the interval of 12.1% to 9.7% (Table 1). The low recovery of OPCs suggests that some parts of them are left bonded to the resin matrix.

Moreover, the recovery of OPCs showed a trend to decrease with the increase of the number of purification batches, showing that the irreversible bonding of OPCs increases with the number of SPE batches. These results demonstrate that the used resin was progressively losing its adsorption capacity.

Finally, the NP-HPLC-PAD analysis of 20 mg/mL solutions prepared from the freeze-dried OPC-rich fractions revealed that the initial high recovery of catechins and OPCs from SPE_1_ decreased slightly after SPE_2_ and drastically with SPE_3_ and SPE_4_ batches, suggesting that the resin was losing not only adsorption capacity but also selectivity of the separation of catechins and OPCs in front of PPCs (Figure 6).

For calculating the quantitative mass balance of the process, only the amounts of species that pass freely through the adsorbent (sugars, sugar alcohols, di- and tricarboxylic acids, minerals and other very polar and ionic species) were taken in consideration. Theoretical recovery of OPCs was calculated by the difference between the amounts of matter entering the resin (per batch) and the amounts of species recovered in the permeates. The results showed that the sugar-rich fraction was more abundant (30%) than OPCs (22.5% based on the extracted whole dry matter; Figure 7).

It is also important to note that all attempts to regenerate the resin with acetone, butanol, NaOH and HCl solutions were not effective. Next, the XAD7HP resin was substituted with XAD16 resin, but the loss of adsorption capacity occurred again after 3–4 loads with the same ultrafiltrate. Kammerer et al. had already warned that some polymeric phenols could attach irreversibly to the active sides of adsorption resin and decrease their adsorption capacity in the purification of phenolics from crude plant extracts [47]. That was one of the reasons to remove the PPC fraction first by UF; nevertheless, the results showed that if only macromolecules were responsible for the resin adsorption inhibition, the already done UF clean-up was not efficient enough. These results also suggest that other adsorbent materials should probably be explored as well.

### 3.4. Evaluation of the Global Purification of OPCs

The global effect of purification of OPCs from GSE by both UF and SPE was also assessed by NP–HPLC–PAD, and the obtained results are shown in Figure 8 and Table 2. For simplicity, only the chromatogram of the SPE_1_ fraction is shown in Figure 8.

The global effect of purification of OPCs from GSE by the integrated UF/SPE process allowed a general enrichment of all groups of catechins and OPCs (Figure 8, Table 2). This enrichment was higher for the low molecular mass species (catechins and procyanidin dimers), which reached values of 9.6- to 7.8-fold. For the rest of the studied OPCs, the enrichment decreased progressively with the increase of the molecular mass of the procyanidin oligomers. Regarding PPCs, even though they were reduced to more than half of their initial content in the first SPE_1_ fraction, their proportion increased with each SPE batch due to the loss of adsorption capacity and selectivity of the resin. Figure 8 also shows an overall elevation of the baseline of the HPLC chromatogram of the purified OPC fraction with respect to the clarified GSE, which indicates that there are other molecular species coeluting under the OPCs that cannot be separated at the used chromatographic conditions. Most probably, these species come from the macromolecular PPC fraction, but it is still a challenge to be explored.

Finally, from the 10 L ultrafiltered GSE, only 24.2 g of purified OPCs (those obtained from SPE_1_ and SPE_2_ batches) met the requirements for quality acceptance. The rest of them had to be repurified with new resin. Nevertheless, the integrated UF/SPE process produced an OPC-rich powder of high purity (83% OPCs). The main impurities of this fraction were gallic acid and some other hydroxybenzoic and hydroxycinnamic acids and flavonols (data not shown). It is also important to note that there was no need for the addition of any kind of drying carrier for further stabilisation of the powder because it was fluid and physically stable by itself.

## 4. Conclusions

The effectiveness of a preparative integrated UF/SPE process for the purification of OPCs and PPCs from crude GSE was studied for the first time. Semiquantitative determination of total OPCs and PPCs was carried out by NP–HPLC, also for the first time.

The UF stage was very efficient in the separation of oligomeric (OPCs) from polymeric procyanidins (PPCs), allowing the treatment of 7.2 L of crude, clarified GSE in less than 3 h, with a filtration flux between 6 and 3.5 L/h.m^2^, at P_TM_ as low as 0.5 bar. An acceptable purification of the macromolecular (polymeric procyanidin-rich) fraction (85%) by diafiltration with water was also possible at the same operating conditions for an additional 3 h. The membrane lost up to 5% of its hydraulic permeability after chemical cleaning, which is also acceptable, making the process scalable to a pilot scale.

The separation of very polar and ionic species of the grape seed extract, such as sugars, sugar alcohols, di- and tricarboxylic acids and minerals from OPCs by preparative column SPE with XAD7HP and XAD16 resins was also very good. However, both adsorbents lost their adsorption capacities quickly and irreversibly, due probably to the retention of OPCs/PPCs. The treatment produced higher enrichment of monomeric flavan-3-ols and OPCs. The highest increase of 9.6- and 7.8-fold was achieved for catechins and procyanidin dimers, respectively. However, the enrichment of OPCs decreased with the increase of the length of the flavanol chain. The obtained results suggest that the use of these resins for further purification of OPCs is possible but very limited and that, probably, other adsorbent materials have to be explored or increased expenses for the restitution of the exhausted resins should be assumed.

The global effect of purification of OPCs from GSE by the integrated UF/SPE process allowed the recovery of 24.2 g of highly purified OPCs (with 83% purity) from 14.4 L of crude grape seed extract. There was no need for the addition of any kind of drying carrier for further stabilisation of the OPC-rich powders because they were fluid and physically stable by themselves.

The quantitative fractionation of the crude grape seed extract by UF/SPE and the use of a semiquantitative NP–HPLC method showed that the main components of the extract were the polymeric procyanidins (47.5%), whereas the oligomeric procyanidins were as much as 22.5%. It is important to note that the crude grape seed extract also contains an important amount (30%) of sugars, sugar alcohols, di- and tricarboxylic acids, minerals and other very polar and ionic species that should be taken into consideration as impurities.

## Figures and Tables

**Figure 1 membranes-10-00147-f001:**
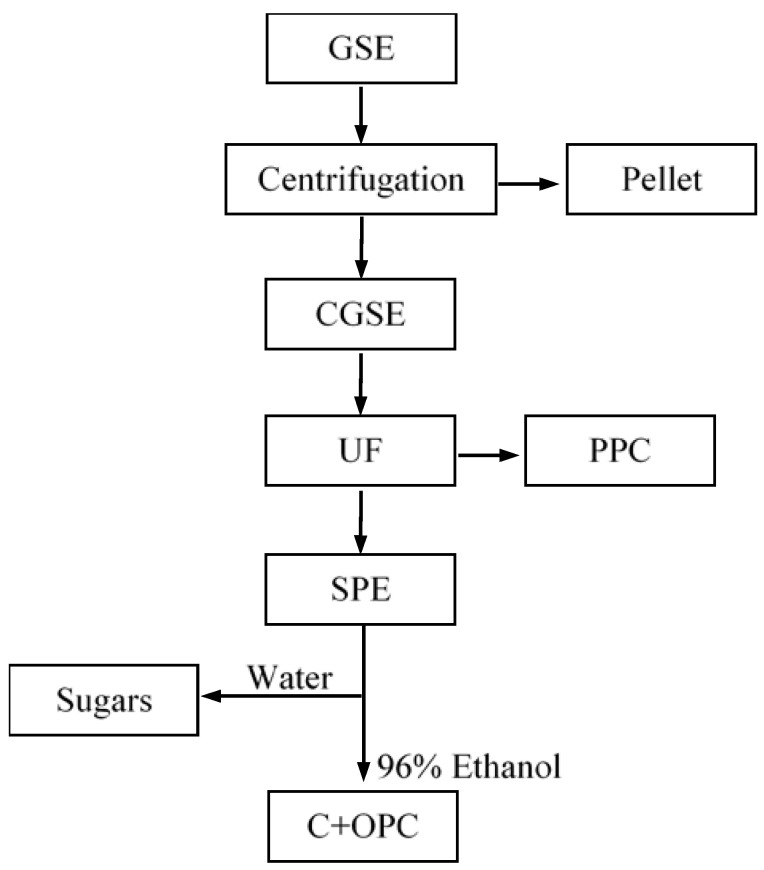
Flow chart of the purification pathway of oligomeric procyanidins from grape seed extract (GSE—grape seed extract, CGSE—clarified GSE, UF—ultrafiltration, SPE—solid phase extraction, PPC—polymeric procyanidin-rich fraction, C—catechins (monomeric flan-3-ols), OPC—oligomeric procyanidin-rich fraction, include some hydroxybenzoic and hydroxycinnamic acids and flavonols, Sugars—sugar-rich fraction, also including sugar alcohols, di- and tricarboxylic acids, minerals, and other very polar and ionic species).

**Figure 2 membranes-10-00147-f002:**
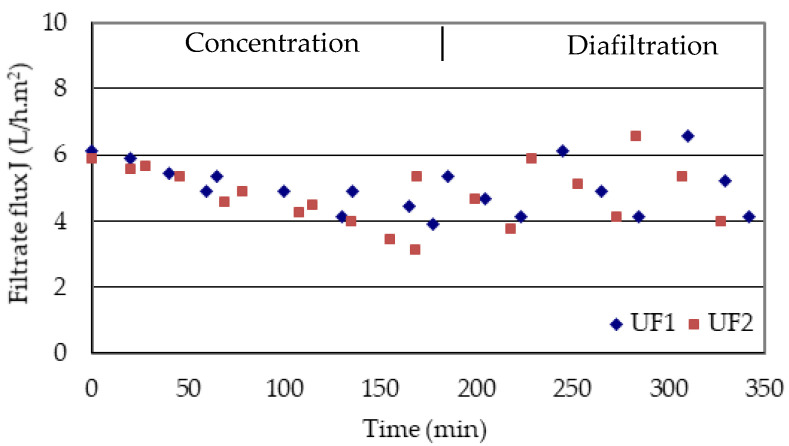
Permeate flux kinetics during UF and diafiltration of both replicates (UF_1_ and UF_2_) of the studied grape seed extract (GSE) by the 10-kDa MMCO membrane (P_TM_ 0.5 bar, temperature 15–26 °C).

**Figure 3 membranes-10-00147-f003:**
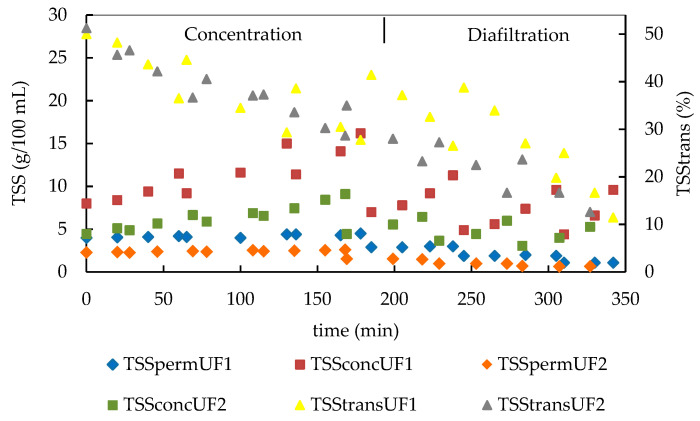
Distribution of total soluble substances (TSSs) in the permeate and concentrate streams and transference of TSSs through the membrane of both replicates (UF_1_ and UF_2_) of the studied GSE during UF and diafiltration by a 10-kDa MMCO membrane.

**Figure 4 membranes-10-00147-f004:**
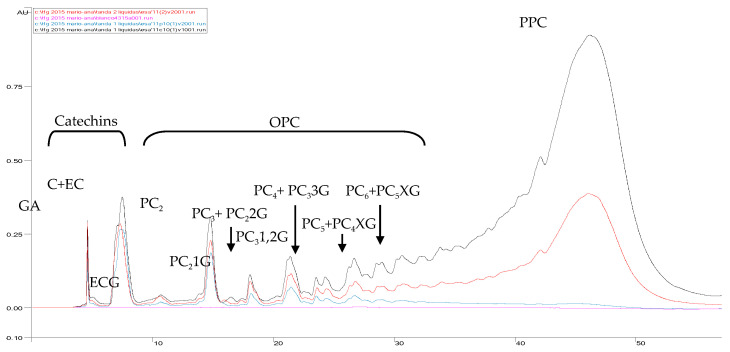
Overlay of NP–HPLC–PAD chromatograms (280 nm) of the initial (clarified) grape seed extract (GSE; red), the final permeate (blue), the final concentrate (black) and the HPLC baseline (violet) after GSE fractionation by a 10-kDa MMCO membrane (data refer to one replicate). GA—gallic acid, C+EC—catechin and epicatechin, ECG—epicatechin gallate, PC_2_ to PC_6_—procyanidin dimers to hexamers, PC_2_1G—monogaloylated procyanidin dimer, PC_2_2G—digaloylated procyanidin dimer, PC_3_1,2G—mono- and digaloylated procyanidin trimers, PC_3_3G—trigaloylated procyanidin trimer, PC_4_XG and PC_5_XG—multigaloylated procyanidin tetramers and pentamers, OPC—oligomeric procyanidins, PPC—polymeric procyanidins.

**Figure 5 membranes-10-00147-f005:**
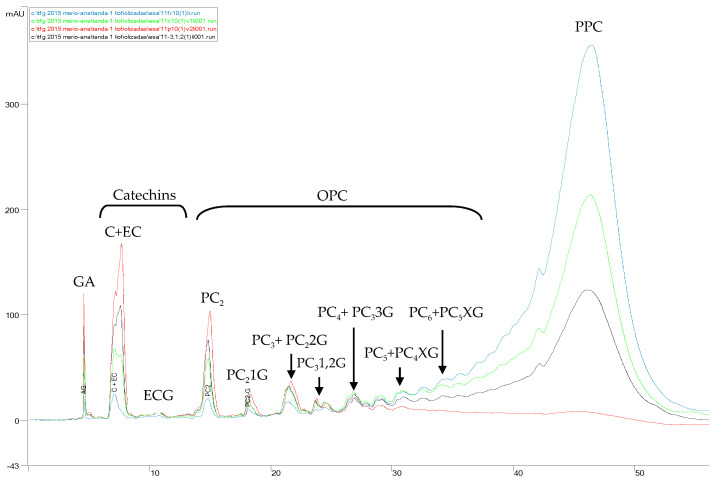
Overlay of the NP–HPLC-PAD chromatograms (280 nm) of the initial (clarified) grape seed extract (GSE; black), the whole permeate (red), the concentrate (green) and the diafiltered concentrate (blue) after GSE fractionation by the 10-kDa MMCO membrane (data refer to one replicate). GA—gallic acid, C+EC—catechin and epicatechin, ECG—epicatechin gallate, PC_2_ to PC_6_—procyanidin dimers to hexamers, PC_2_1G—monogaloylated procyanidin dimer, PC_2_2G—digaloylated procyanidin dimer, PC_3_1,2G—mono- and digaloylated procyanidin trimers, PC_3_3G—trigaloylated procyanidin trimer, PC_4_XG and PC_5_XG—multigaloylated procyanidin tetramers and pentamers, OPC—oligomeric procyanidins, PPC—polymeric procyanidins.

**Figure 6 membranes-10-00147-f006:**
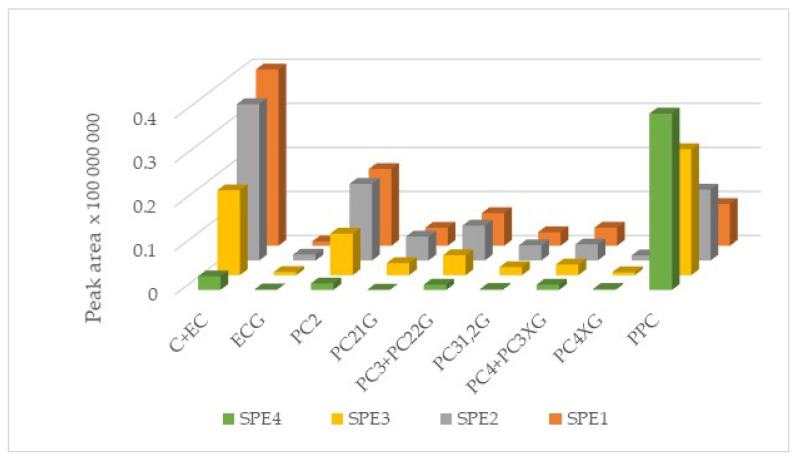
Areas of the peaks (acquired by NP–HPLC-PAD at 280 nm) corresponding to monomeric nongalloylated (catechin and epicatechin, C+EC) and galloylated (epicatechin gallate, ECG) catechins to tetrameric multigalloylated (PC_4_XG) and polymeric procyanidins (PPCs) from the four SPE batches of the GSE ultrafiltrate.

**Figure 7 membranes-10-00147-f007:**
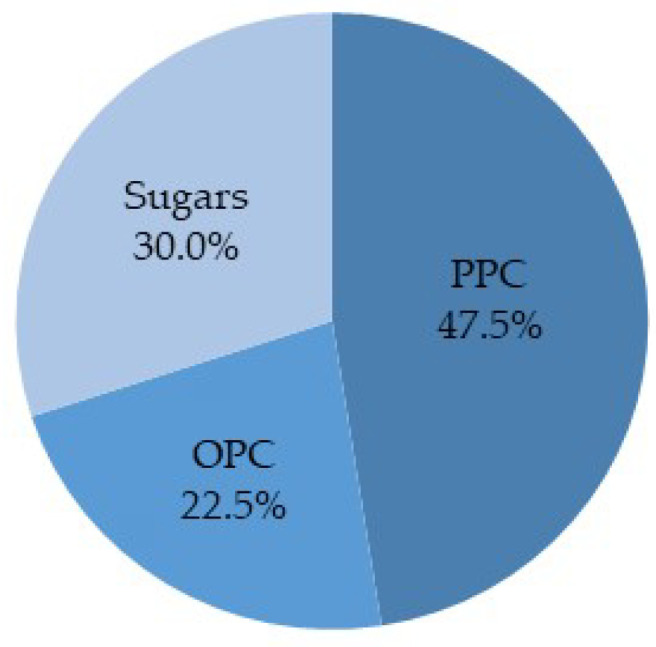
Relative theoretical amounts of the main grape seed components after fractionation by the integrated UF/SPE process.

**Figure 8 membranes-10-00147-f008:**
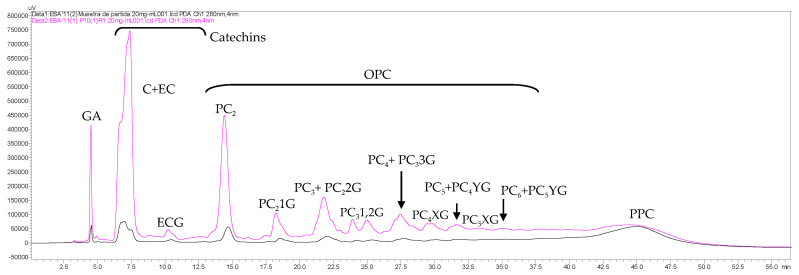
Overlay of NP–HPLC–PAD chromatograms (280 nm) of 20 mg/mL reconstituted initial (clarified) grape seed extract (GSE; black) and 20 mg/mL reconstituted OPC-rich fraction (after UF and SPE_1_) (magenta) acquired at 280 nm. GA—gallic acid, C+EC—catechin and epicatechin, ECG—epicatechin gallate, PC_2_ to PC_6_—procyanidin dimers to hexamers, PC_2_1Gmonogaloylated procyanidin dimer, PC_2_2G—digaloylated procyanidin dimer, PC_3_1,2G—mono- and digaloylated procyanidin trimers, PC_3_3G—trigaloylated procyanidin trimer, PC_4_XG and PC_5_XG—galoylated procyanidin tetramers and pentamers of low degree of galloylation, PC_4_YG and PC_5_YG—galoylated procyanidin tetramers and pentamers of high degree of galloylation, OPC—oligomeric procyanidins, PPC—polymeric procyanidins.

**Table 1 membranes-10-00147-t001:** Total soluble substances per batch (TSS/batch), total soluble substances recovered in the permeate (TSS_perm_), total soluble substances recovered in the permeate and referred to 100 g of TTS/batch (TSSR_perm_), total soluble substances recovered in the OPC fraction (TSS_OPC_) and total soluble substances recovered in the OPC fraction and referred to 100 g of TTS/batch (TSSR_OPC_) after solid-phase extraction (SPE) of the ultrafiltered GSE permeate.

SPE_batch_	TSS/batch	TSS_perm_	TSSR_perm_	TSS_OPC_	TSSR_OPC_
	(g)	(g)	(%)	(g)	(%)
SPE_1_	105	58.9	56.1	12.7	12.1
SPE_2_	105	59.5	56.7	11.5	10.9
SPE_3_	105	60.3	57.4	10.9	10.4
SPE_4_	105	59.3	56.5	10.2	9.7

**Table 2 membranes-10-00147-t002:** Flavan-3-ol enrichment (by groups) in purified OPC fractions after UF and after each of the four consecutive SPE batches.

Flavan-3-ols	Flavan-3-ol Enrichment (Folds)			
	SPE_1_	SPE_2_	SPE_3_	SPE_4_
C+EC	9.6	8.5	4.6	0.7
ECG	4.0	5.4	2.9	0.4
PC_2_	7.8	7.8	4.2	0.6
PC_2_1G	5.6	7.4	3.7	0.1
PC_3_+PC_2_2G	5.7	6.1	3.5	0.9
PC_3_1,2G	5.7	6.3	3.4	0.4
PC_4_+PC_3_3G	5.3	4.7	3.1	1.5
PC_4_XG	5.9	5.0	3.0	0.9
PC_5_+PC_4_YG	6.2	5.0	4.5	4.0
PC_5_XG	5.3	4.4	3.1	1.9
PC_6_+PC_5_YG	3.3	2.7	2.1	4.1
PPC	0.9	1.4	2.4	3.5

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
