# Peer review of "Evaluation of an Integrated Ultrafiltration/Solid Phase Extraction Process for Purification of Oligomeric Grape Seed Procyanidins"

_membranes, 2020, doi:10.3390/membranes10070147_

Round 1
Reviewer 1 Report
Review Membranes 834861
Interesting and well-written paper. The authors are requested to address the following comments/suggestions.
The rationale for selecting a regenerated cellulose based UF membrane with nominal MWCO of 10 kDa is not entirely clear. The authors are encouraged to motivate this membrane choice in view of the envisaged fractionation, and perhaps refer to previous experience or literature studies.
Also, the experimental results in regard of suboptimal purity-yield trade-off (cut-off curve not really sharp under conditions studied) should be discussed in terms of the properties of this membrane and the applied operating conditions (e.g. low TMP).
Please describe the geometry of the spiral-wound UF membranes: dimensions, feed spacer, etc. What was the feed flow, cross-flow velocity and pressure difference over the membrane?
During the concentration, fresh extract was being dosed, but it is unclear how this was done practically. On constant volume basis, or otherwise? Using a dosing pump, or manually?
The temperature was not kept constant and varied significantly (15-26°C, see caption Figure 2). How does this impact on the separation, especially the sharpness of the effective cut-off between OPC and PPC?
The authors state that the UF concentration-purification process is “proper for scaling up to pilot scale”, but it is not clear on what this is based. Is this substantiated by a positive techno-economic evaluation with the obtained yield and purity? Or does it simply refer to the scalability of the membrane process itself?
The word “transformation” in the first sentence of the conclusions section sounds weird. Also, the limited yield should be discussed in the conclusion, rather than just stating “UF was very efficient”.
Reviewer 2 Report
The manuscript investigated an integrated ultrafiltration/solid phase extraction process for purification of oligomeric grape seed procyanidins. This research was generally innovative. However, considering the detailed investigation work, the work may provide some useful information for other readers. My opinion is to be accepted for publication in the journal after major revision. (1) Abstract: These sentences are not clear for me to understand the investigation. This is the most important and widely read part of your paper. It should contain all the important elements of your paper, and nothing else. From the abstract it is still not clear to me what exactly was the novelty of the study. So, to me, the Abstract should begin very general then gradually narrow to the topic, then end general. (2) Conclusions: Conclusion needs to be expressed in summative language. In my opinion, this part should be improved.Author Response
Please see the attachment file

Round 2
Reviewer 2 Report
I have no comments on the mansucript and the manuscript can be accepted for publisition by the journal now.